# Understanding Diversity: The Cultural Knowledge Profile of Nurses Prior to Transcultural Education in Light of a Triangulated Study Based on the Giger and Davidhizar Model

**DOI:** 10.3390/healthcare13151907

**Published:** 2025-08-05

**Authors:** Małgorzata Lesińska-Sawicka, Alina Roszak

**Affiliations:** Department of Nursing, Academy of Applied Sciences in Konin, 62-510 Konin, Poland; alina.roszak@konin.edu.pl

**Keywords:** cultural competence, nursing, Giger and Davidhizar model, content analysis, triangulation, transcultural education

## Abstract

**Introduction:** The increasing cultural diversity of patients poses new challenges for nurses. Cultural competence, especially knowledge of the cultural determinants of health and illness, is an important element of professionalism in nursing care. The aim of this study was to analyse nurses’ self-assessment of cultural knowledge, with a focus on the six dimensions of the Giger and Davidhizar model, prior to formal training in this area. **Methods:** A triangulation method combining qualitative and quantitative analysis was used. The analysis included 353 statements from 36 master’s student nurses. Data were coded according to six cultural phenomena: biological factors, communication, space, time, social structure, and environmental control. Content analysis, ANOVA, Spearman’s rank correlation, and cluster analysis (k-means) were conducted. **Results:** The most frequently identified that categories were environmental control (34%), communication (20%), and social structure (16%). Significant knowledge gaps were identified in the areas of non-verbal communication, biological differences, and understanding space in a cultural context. Three cultural knowledge profiles of the female participants were distinguished: pragmatic, socio-reflective, and critical–experiential. **Conclusions:** The cultural knowledge of the participants was fragmented and simplified. The results indicate the need to personalise cultural learning and to take into account nurses’ level of readiness and experience profile. The study highlights the importance of the systematic development of reflective and contextual cultural knowledge as a foundation for competent care.

## 1. Introduction

Contemporary nursing presents professionals with new challenges resulting from the growing cultural diversity of patients. Globalisation and migration mean that nurses are increasingly caring for people of different ethnic, linguistic, religious, and social backgrounds. Effective healthcare requires understanding and respecting the cultural specificity of the individual, which affects the quality of services provided and patient satisfaction. In recent years, both the scientific literature and international organisations such as the WHO have consistently emphasised the importance of cultural competence in nursing. Nurses with a higher level of this competence communicate more effectively with patients, which leads to better health outcomes, faster recovery, and reduced communication barriers [1,2,3,4,5].

The concept of cultural competence was introduced by Madeleine Leininger in the 1970s [6] and has evolved since then. Cultural competence is not an innate trait but an acquired process that requires conscious learning, reflection, and professional experience [7]. The American Academy of Nursing has defined it as the acquisition of knowledge, understanding, and skills about different cultural groups that enable the provision of culturally acceptable care [8]. The concept of culture does not refer exclusively to ethnicity, cultural, or religious groups but also to everyday aspects of functioning in different environments, systems, and organisational structures, such as gender, clothing, gestures, cooking, the meaning of food, perception of reality, thoughts, behaviours, attitudes, and others [9].

This study focuses on cultural competence understood as a multidimensional ability, comprising cognitive (knowledge), affective (attitudes), and behavioural (skills) components, with particular emphasis on the knowledge component, which forms the basis for the development of the other dimensions.

The development of cultural competence is an acquired process, requiring conscious learning, reflection, and experience. It evolves with the development of self-awareness, contacts with people from different cultures, and professional practice [10,11]. Various theoretical models have been developed to support this process in nursing, including the Madeleine Leininger model [12]; the Purnell model [13]; the Campinha–Bacote model [14]; and the Papadopoulos, Tilki, and Taylor model of developing cultural competence [15]. Although each of these models offers a valuable framework for understanding and developing cultural competence, they differ in scope, complexity, and emphasis on particular aspects. For example, Leininger’s model (Cultural Care Theory) focuses on transcultural nursing as a field that aims to provide culturally appropriate care, using concepts such as cultural care behaviour and negotiation [12]. Purnell’s model, on the other hand, offers a comprehensive and holistic framework for assessing the cultural characteristics of individuals, families, and communities, taking into account twelve cultural domains, from heritage to health perception [13]. The Campinha–Bacote model emphasises the process by which nurses become culturally competent through five constructs: cultural awareness, cultural knowledge, cultural skills, cultural encounters, and cultural desire [14]. The Papadopoulos, Tilki, and Taylor model focuses on the process of developing competence through four stages: cultural awareness, cultural knowledge, cultural sensitivity, and cultural competence [15].

The study presented here uses the Transcultural Assessment Model, developed in 1995 by Giger and Davidhizar, which distinguishes six cultural phenomena relevant to nursing care: communication, space, time, environmental control, social structure, and biological factors. The choice of this model was based on several key advantages that make it particularly suitable for assessing the knowledge component among Polish nursing students at the master’s level who have not yet undergone formal cross-cultural training. Firstly, the Giger and Davidhizar model is valued for its simplicity and clarity. Its structure is based on six clearly defined cultural phenomena: communication, space, time, environmental control, social structure, and biological factors. This clear categorisation facilitates the study and analysis of specific areas of cultural knowledge, which was crucial for the purpose of this study—to identify gaps in knowledge prior to formal training. Secondly, the model has high practical utility in nursing education and clinical case analysis. Its specific dimensions allow for the effective design of clinical scenarios, which was used in the research tool (Appendix A), enabling students to relate to real-life situations and demonstrate their knowledge in specific contexts. Unlike more abstract or process-oriented models, Giger and Davidhizar provide specific categories that can be directly related to patient interaction and the assessment of their cultural needs. This makes it a particularly effective tool for analysing self-assessment of knowledge, as respondents can more easily categorise their thoughts and observations. It enables the identification and consideration of cultural differences in nursing practice [16]:Communication is a continuous process in which one person can interact with another through written or spoken language, gestures, facial expressions, body language, space, or other symbols.Space refers to the distance and techniques of showing proximity used in verbal and non-verbal interactions. All communication takes place in the context of space.Time is a very important aspect of interpersonal communication. Cultural groups may be oriented towards the past, present or future. Cultures refer to time in terms of clock time or social time—some groups function on the basis of social time.Environmental control refers to a person’s ability to control nature and to plan and direct factors in the environment that affect them.Social structure is the way in which a cultural group organises itself around a family or community.Biological differences are specifically genetic conditions, patterns of growth and development, functioning of body systems, anatomical characteristics of race, skin and hair physiology, incidence of disease, and resistance to disease.

The model is characterised by simplicity, clarity, and high practical utility and is therefore particularly valued in nursing education and clinical case analysis. It enables the identification of cultural differences and the assessment of nurses’ preparedness to work with culturally diverse patients. Consequently, nurses should not only be aware of cultural differences but also be able to adapt their actions to the specific needs of patients, which is key to providing holistic healthcare [17]. Furthermore, the integration of cultural education into nursing curricula is essential to prepare nurses to work in a diverse environment. Appropriate training allows nurses to develop the skills needed to deal effectively with cultural differences and promote health in multicultural communities [18].

The aim of the study was to analyse nurses’ knowledge of selected cultural phenomena relevant to nursing care, with reference to the Giger and Davidhizar model.

The following PICO-type research question was formulated:-P (Population): nurses who are master’s students in Poland with professional experience.-I (Intervention): analysis of self-assessed cultural knowledge in relation to the six dimensions of the Giger and Davidhizar model.-C (Comparison): a lack of formal cultural preparation (no completed subject ‘Nursing in a multicultural environment’ or postgraduate courses on a similar topic).-O (Outcome): the identification of their level of cultural knowledge and deficit areas.

Which areas of cultural knowledge, according to Giger and Davidhizar’s model, are least represented in the self-assessment of nurses (MSc students) who have not yet received formal cultural training?

Therefore, the research question is as follows: what is the scope and nature of cultural knowledge among master’s degree nursing students assessed prior to a course in transcultural nursing based on the Giger and Davidhizar model?

## 2. Materials and Methods

### 2.1. Design

The study employed a descriptive research design using methodological triangulation, understood here as method triangulation—the use of both qualitative and quantitative research methods to study the same phenomenon. The aim of this approach was to provide a broader, more nuanced understanding of nurses’ cultural knowledge and to enhance the validity and trustworthiness of the findings.

The qualitative part consisted of a structured content analysis based on a deductive approach using Giger and Davidhizar’s model as a theoretical framework. It allowed in-depth exploration of how nurses formulated their knowledge in narrative form, identifying not only the presence of particular cultural concepts but also their contextual and interpretative meanings.

The quantitative part complemented this by enabling the systematic categorisation and measurement of the frequency and distribution of these cultural dimensions as well as identifying relationships and profiles through statistical analyses (e.g., ANOVA, correlation, cluster analysis). The combination of both approaches allowed for the convergence of findings, reinforcing the internal validity of the results and enabling the identification of knowledge gaps that may not have been apparent through one method alone.

The integration of both methods was sequential and complementary: the qualitative content analysis served as the primary analytic strategy, while the quantitative component offered statistical verification and differentiation of patterns identified in the qualitative data. This interaction enabled both a depth of interpretation and empirical generalisability within the study’s scope.

### 2.2. Sample and Setting

The study was conducted between 2022 and 2024 at the Academy of Applied Sciences (ANS) in Piła and the ANS in Konin. The education of nurses in Poland is based on a national training standard that requires a two-year (four-semester) master’s degree programme following a bachelor’s degree in nursing. The research was carried out prior to the implementation of any formal transcultural nursing education.

The sampling strategy was purposive and convenience based. Students were selected intentionally based on their status as second-degree (master’s level) nursing students with ongoing clinical practice and no prior formal training in transcultural healthcare. Inclusion criteria for the study included the following: status as a student of nursing (second-level studies), current license to practise as a nurse, and voluntary consent to participate in the study. Exclusion criteria included the following: participation in a pilot study or attendance at a postgraduate course on “Intercultural Dialogue in Healthcare”. Recruitment was carried out with the help of the supervisors of each year group, who were not involved in the study. No financial or academic incentives were offered to respondents.

Of the 54 students invited, 36 agreed to participate (response rate: 66.7%). Although the sample size may appear limited, it was considered sufficient for the qualitative content analysis and exploratory quantitative procedures used in the study (e.g., clustering). This sample size enabled the identification of variation and patterns in cultural knowledge across key categories. Given the qualitative–exploratory character of the study, no power calculation was performed. The majority of participants were women (n = 33; 91.67%), which is consistent with the gender distribution in the general nursing population in Poland. Respondents were aged between 23 and 54 years (M = 33.61, SD = 3.39) and worked in hospital wards such as paediatric, internal medicine, neurology, nephrology, lung diseases, and hospital emergency department (n = 31; 86.11%) and in primary care (n = 5; 13.89%). Length of service as a nurse ranged from 0.5 years to 30 years (M = 18.3, SD = 3.10). In their professional work, 58.7% of respondents had previous contact with people from other cultures, mainly Roma, Ukrainians, Germans, and English. None of the respondents had worked abroad, and 15.5% were considering working abroad in the future.

### 2.3. Procedure of the Study

The idea for the study was born during the postgraduate course “Intercultural Dialogue in Healthcare”, which analysed different clinical scenarios based on the Giger and Davidhizar Transcultural Assessment Model. Discussions with the course participants revealed significant differences in the level of knowledge regarding the different cultural dimensions, which inspired the design of the research tool and an in-depth analysis of the nurses’ cultural knowledge.

Prior to the main study, a pilot study was conducted to assess the comprehensibility of the tool and to initially verify its reliability. The pilot study involved 10 nurses meeting the inclusion criteria who were not included in the main study. In parallel, the understanding of the scenarios and questions, as well as their relevance to the dimensions of the Giger and Davidhizar model, was verified in a group of female participants of the aforementioned course. All cases were discussed together, and modifications were made based on the suggestions made, primarily in the description of the ‘social structure’ category.

In the main survey, prior to the start of classes in the semester in which the course ‘Multicultural Nursing’ was scheduled, students were asked by their group supervisors to participate in the survey. Participants were informed of the purpose and nature of the study, how to respond, and that they could withdraw from participation without any consequences. Before the study began, participants verbally agreed to participate. They were then given a sheet with questions and instructions for participating in the study. Written responses were given at the university and took 45–60 min to complete.

### 2.4. Instrument

The research tool was a self-administered questionnaire with 6 case descriptions corresponding to the cultural phenomena identified by Giger and Davidhizar. Two questions were prepared for each description: one on expertise and one on proposed nursing interventions (Appendix A). To establish content validity, the tool underwent expert review by three experienced nursing educators specialising in transcultural care. These experts assessed the relevance and clarity of the case scenarios and questions. Based on their feedback, minor adjustments were made, particularly in the description and interpretation of the “social structure” dimension. In the present study, only questions on expertise were used for analysis, as the aim was to assess the participants’ level of cultural knowledge, not practical skills. The decision to analyse only the “expertise” questions was based on the study’s primary objective: to evaluate the cognitive component (knowledge) of cultural competence prior to formal education. Intervention questions, which required the formulation of culturally appropriate care strategies, engage the behavioural dimension and were considered beyond the intended scope of this phase. They are planned for analysis in subsequent studies.

### 2.5. Data Analysis

A deductive content analysis was conducted using Giger and Davidhizar’s Transcultural Assessment Model as an analytical framework. The analysis followed the stages described by Elo and Kyngäs (Figure 1) [19].

The analysis of the codes was carried out by two researchers independently of each other, thus reducing the risk of interpretation error and increasing the reliability of the results. All coding discrepancies were discussed by the researchers until consensus was reached.

Qualitative analysis was carried out using MAXQDA24, (version 24.5.1; VERBI Software, Berlin, Germany) which enabled structured coding, graphical presentation (e.g., code matrices), and statistical summary of code frequencies. The coding system adopted in the study is shown in Table 1.

In order to increase the reliability of the coding process, the inter-coder agreement coefficient—Cohen’s Kappa—was calculated. The resulting κ = 0.62 (Z = 12.70; *p* < 0.000001) indicates significant inter-encoder agreement, as interpreted by Landis and Koch [20]. In addition, methodological triangulation was used to strengthen the reliability of the results by including quantitative methods. These included code frequency analysis, analysis of variance (ANOVA), Spearman rank correlation, and k-means cluster analysis. The quantitative techniques used were applied to statistically verify relationships, explore variation in cultural knowledge dimensions, and identify distinct knowledge profiles. These methods directly correspond to the central research question regarding which areas of cultural knowledge are underrepresented among MSc nursing students prior to formal training. A statistical significance level of *p* = 0.05 was adopted.

### 2.6. Statement of the Ethics Committee

The study was conducted in accordance with the principles of the Declaration of Helsinki and the applicable standards for the conduct of scientific research—the Act of 5 December 1996 on the Professions of Physician and Dentist (Journal of Laws 2023, item 1516). Due to the nature of the study—an anonymous survey not involving medical intervention—no approval from the bioethics committee was required. All participants were fully informed about the purpose and procedures of the study. Written informed consent was obtained from each participant. Participation was voluntary and anonymous. Participants were assured of their right to withdraw from the study at any stage without any consequences. The data were stored securely, with no identifying information collected. Only the research team had access to the anonymised responses.

## 3. Results

The study analysed a total of 353 text segments, assigned to six cultural dimensions according to Giger and Davidhizar’s model. The most frequent categories were environmental control (33.99%), communication (19.55%), and social structure (15.58%). Space (12.75%), time (9.92%), and biological factors (8.22%) were less frequently addressed (Table 2).

The detailed analysis showed that within each category, specific codes predominated (Table 3).

For biological factors, the colour of the skin and mucous membranes was indicated most frequently (M = 0.67), while their texture was indicated less frequently (M = 0.14). The high variation in responses (SD = 0.667) confirms the uneven level of knowledge in this area. The communication category was dominated by the code ‘lack of language knowledge’ (M = 0.89; SD = 0.314), significantly outperforming other aspects such as a lack of knowledge of non-verbal communication (M = 0.08), a lack of understanding of medical terminology (M = 0.19), or fear of the unknown (M = 0.56). In the space dimension, the most frequent code was ‘lack of privacy’ (M = 0.69), while aspects such as dependence on staff (M = 0.31) or lack of control over one’s own behaviour (M = 0.25) received less attention. For the social structure category, shared values, symbols, and behaviours (M = 0.67) and a sense of community (M = 0.64) were rated highest. Tradition was mentioned infrequently (M = 0.22). In the category of time, the belief that it was culturally conditioned predominated (M = 0.94), while responses indicating the absence of this dependence were marginal (M = 0.03). In the category of environmental control, belief in the forces of nature and God (M = 0.75), a lack of trust in the healthcare system (mean = 0.61), and a low level of health knowledge (M = 0.56) were most frequently indicated. The high standard deviations (up to SD = 0.550) indicate a large individual variation in the perception of this category (Table 3).

Figure 2 presents the distribution of responses within the six cultural dimensions. The most frequent references concerned communication and space, while the least frequent were related to time and social structure. Each row is a code, each column is a respondent, and a square indicates an individual’s assignment of a particular code. The more prominent the density of the code, the greater the number of indications of it, e.g., the density of codes in the communication dimension (particularly ‘lack of language skills’) contrasts with the sparseness of coding in the biological factors dimension, visualising key knowledge disparities.

### Statistical Analyses

One-way analysis of variance (ANOVA) showed statistically significant differences in the frequency of code assignments between cultural categories (F = 41.44; *p* < 0.000001).

Spearman’s rank correlation analysis revealed significant positive correlations (Table 4):-Between space and environmental control (ρ = 0.50);-Between social structure and environmental control (ρ = 0.36);-Between social structure and space (ρ = 0.38).

A weak or negative correlation occurred between time and communication (ρ = −0.24). Biological factors showed very weak correlations with the other categories.

The number of clusters was determined using the elbow method, which showed that a three-cluster solution best balanced interpretability and explained variance. The k-means method was chosen due to its efficiency and suitability for small, non-hierarchical datasets.

Cluster analysis (k-means) identified three distinct profiles of study participants (Figure 3):-Cluster 0—“low perception of cultural barriers, pragmatic profile” (n = 17). This was characterised by the lowest scores in all areas (0.1–0.6 codes on average), with no dominant categories. The highest number of codes appeared in the areas: communication and social structure.-Cluster 1—“profile with high social and communication sensitivity” (n = 12). Respondents in this group scored highest in categories such as communication (average 2.08 assignments), time (1.33), social structure (3.17), and biological factors (1.00). They were characterised by a high awareness of cultural differences related to interpersonal relationships, language, religion, and community influence. This is the group of nurses most sensitive to cultural determinants of care.-Cluster 2—“critical and experienced profile” (n = 13). The highest scores here were in the categories of communication (2.54), social structure (4.00), biological factors (1.31), and social structure (1.15). Respondents in this cluster strongly emphasised specific barriers—a lack of trust, patients’ religiosity, language difficulties, and the social situation.

Figure 2 displays the cluster solution, showing three distinct profiles of cultural knowledge among participants.

The most differentiating codes of the individual clusters were as follows:Communication: a lack of language skills—significantly higher in cluster 2, indicating great language difficulties in this group.Social structure: belief in the healing powers of nature and God—particularly significant for cluster 2.Communication: a lack of trust—clearly raised in clusters 1 and 2.Social structure: social situation—plays a greater role in clusters 1 and 2 than in 0.Communication: a lack of knowledge of medical phrases—mainly cluster 2.Time: the dependence of time on culture—much more emphasised in cluster 1.Social structure: the organisation of family life—clearly important for cluster 2.Social structure: experiences of other community and family members—key in cluster 2.Communication: fear of the unknown—more frequent in cluster 1.Biological factors: the colour of skin and mucous membranes—more frequent in cluster 1 and 2.

Table 5 provide an overview of the identified cultural knowledge profiles, including defining characteristics and response patterns.

## 4. Discussion

Although the general level of cultural competence in nursing has been studied many times, few studies have focused on identifying specific knowledge profiles and structural links between different dimensions of culture, especially in countries with relatively low ethnic diversity [21,22,23], such as Poland. The results of the study revealed significant differences in nurses’ perceptions and understanding of specific cultural dimensions. In particular, categories such as environmental control, communication, and social structure were most frequently identified, while biological factors, space, and time appeared much less frequently. Such a distribution may indicate a greater ‘visibility’ of certain categories in daily clinical practice and thus their greater recognition.

Identified deficits in cultural knowledge were particularly focused on non-verbal communication, the understanding of medical terminology, and biological aspects of differences. The reduction of communication to a language barrier may reflect a simplistic understanding of the phenomenon. Meanwhile, the literature emphasises that it is non-verbal communication, including gestures, eye contact, body posture, or interpersonal distance, that plays a key role in intercultural relations [24,25]. Ignoring it may lead to a misunderstanding of the patient’s needs, especially in cultures where non-verbal communication has a dominant function [26].

The quantitative and qualitative data also allowed for an analysis of the associations between the different categories. Statistically significant correlations, such as the relationship between space and environmental control (ρ = 0.50), suggest that for respondents, the patient’s need for privacy was strongly related to their sense of control over their environment and the treatment process. In contrast, the almost complete lack of correlation of biological factors with other dimensions (ρ close to 0) may indicate their isolated, purely medical perception—devoid of a broader cultural or communicative context. This is supported by the observations of other researchers that nurses use skin assessment standards developed for fair-skinned patients, which can lead to diagnostic errors and inequalities in care [27,28].

The results of the cluster analysis (clusters) brought a new interpretative dimension to the assessment of nurses’ cultural knowledge profiles. Cluster 0 (“low perception of cultural barriers, pragmatic profile”) comprised the largest group of respondents. It was characterised by low values in all dimensions of the Giger and Davidhizar model, with no dominant categories. This may reflect a general but superficial understanding of culture in the context of care, with limited practical experience. Cluster 1 (“profile with high social and communication sensitivity”), showed the highest mean code assignments in the areas of communication, time, environmental control, and biological factors. Respondents in this group appear to be more reflective and open to the multidimensional aspects of cultural differences, in particular social relationships and trust. Cluster 2 (“critical and experienced profile”) was clearly focused on real barriers to care: language problems, a lack of trust in the health system, patients’ religiosity, and social structure. This profile may reflect professional experiences gained in dealing with patients from groups with strongly held beliefs and traditions. The present analysis revealed that each profile represents a unique way of perceiving culture in nursing practice rather than just quantitative differences in code assignments. Such variation points to the need for an individualised approach in cultural education tailored to the level of awareness, type of experience, and reflective readiness of each group of nurses. The results of the cluster analysis are also worth interpreting in the context of the specificity of the healthcare system and social structure in Poland. The ethnically homogeneous nature of Polish society, with a limited number of cultural minorities in daily nursing practice, may contribute to the fact that nurses‘ cultural knowledge is often more rooted in theoretical frameworks and academic learning rather than extensive, direct clinical experience with diverse populations. This observation is supported by the identified knowledge profiles, particularly ‘Cluster 0—low perception of cultural barriers, pragmatic profile’, which demonstrated a general but superficial understanding of cultural dimensions. This suggests that without consistent exposure, nurses may develop a conceptual understanding of cultural competence without fully grasping its practical nuances. The lack of systematic contact with patients from different cultural backgrounds can indeed limit the opportunity to develop robust cultural competence in practise. At the same time, the particular prominence of categories such as environmental control (e.g., belief in higher forces) and social structure (e.g., influence of community, family) may be the result of observing specific groups in Polish society, such as the Roma community or patients with strong religious affiliations. In these cases, nurses may encounter clear manifestations of a different approach to health and illness, which is reflected in the attributed codes.

The use of a cluster analysis to identify cultural knowledge profiles represents a relatively novel approach in nursing education research. Similar methodological strategies also appear in publications by other researchers [29,30]. The present results, through the extraction of three profiles, confirm that profiling makes it possible not only to identify levels of competence but also to reveal deeper differences in perceptions of culture in the context of healthcare. The practical translation of the extracted profiles indicates that the results of the cluster analysis can become a direct tool in the design of educational programmes. Profiling, as the analysis reveals, suggests the need to individualise the approach; participants with a pragmatic profile (Cluster 0) may need a basic introduction to cultural issues and the development of reflexivity, while groups with high social sensitivity and experienced barriers (Clusters 1 and 2) require more advanced, specialised cultural education.

Additionally, the study in question confirms that certain dimensions of culture, such as non-verbal communication and biological aspects, remain marginalised in nurses’ self-assessments. These findings suggest that competences regarding the nuances of communication, e.g., gestures, eye contact, or body posture, are treated in a simplistic way—limiting them to the language barrier only. The findings highlight that underestimating the importance of non-verbal communication can lead to an inaccurate assessment of patient needs, which is particularly problematic in contexts where this type of communication plays a key role [24,25].

Although the study did not statistically analyse variables such as place of work (e.g., hospital emergency department vs. primary care) or job tenure, these may significantly influence the extent and depth of cultural knowledge. These hypotheses merit further research, preferably using a larger sample and analysing the relationship between knowledge profiles and participants’ demographic and professional characteristics. The identification of diverse participant profiles suggests that the educational approach should be flexible and adapted to the students’ level of knowledge and cultural readiness. Respondents with high social sensitivity may need in-depth expertise, while those with a balanced but superficial profile may need support in developing reflexivity and specific competence. These findings provide valuable insights for the design of nursing education programmes oriented to the real needs and gaps in cultural clinical readiness.

The findings from this study, while rooted in the Polish context, offer broader implications for global nursing education, especially in countries that may share similar demographic characteristics or are experiencing increasing cultural diversification. The identification of distinct cultural knowledge profiles among nursing students underscores the universal need for adaptive and differentiated educational strategies. This study demonstrates that a ‘one-size-fits-all’ approach to cultural competence education is insufficient. Instead, educational programmes globally could benefit from incorporating initial assessments, such as those derived from the Giger and Davidhizar model, to tailor curricula to the specific strengths and deficits of their student cohorts. For instance, in contexts where exposure to diverse cultures is limited, emphasis might be placed on simulation-based learning and immersive virtual realities to bridge the gap between theoretical knowledge and practical application. Conversely, in highly diverse settings, advanced training might focus on complex intercultural communication and negotiation skills. The ability to profile students’ cultural knowledge, as demonstrated here, provides a valuable framework for designing more effective and responsive transcultural nursing education worldwide, fostering genuinely competent care providers capable of navigating diverse healthcare landscapes.

While this study focused specifically on the cognitive dimension of cultural competence and cultural knowledge, it is important to acknowledge the broader conceptual landscape. Cultural competence alone does not encompass the full complexity of providing respectful and equitable care. Concepts such as cultural safety and cultural humility, both rooted in nursing discourse since the 1970s, emphasise the relational, reflective, and patient-defined nature of culturally safe care. Competence cannot be assessed solely through knowledge, and ultimately, it is the patient who determines whether care is culturally appropriate [31,32]. Thus, while models like Giger and Davidhizar’s help in structuring educational interventions, they should be applied critically, with sensitivity to the dynamic, performative, and context-dependent nature of cultural interaction.

### Limitations of the Study

The study conducted, although it provided valuable data on nurses’ self-assessment of cultural knowledge, had several important methodological and contextual limitations.

Although the inclusion of MSc nursing students with professional experience strengthens the credibility of the study, several limitations should be acknowledged. As the data relied on self-assessment, there is a risk of subjectivity, including potential overestimation or underestimation of cultural knowledge and competence. Furthermore, due to the voluntary nature of participation, selection bias cannot be excluded; students who were already interested in cultural issues may have been more likely to take part. Finally, as the sample was limited to Polish MSc nursing students, the generalizability of the findings to other cultural or educational contexts is constrained.

Next, the study focused only on the cognitive component of cultural competence, omitting the affective (attitudes) and behavioural (skills) components. This scope of analysis limits the possibility to fully assess the cultural competence of the participants and does not allow inferences about their actual performance in clinical practice. To address this in future research, it is recommended to employ a multi-method approach, including observation of clinical interactions, simulations, or structured interviews assessing affective and behavioural aspects of cultural competence. Secondly, the small sample size limits the possibility to generalise quantitative results. However, the aim of the study was primarily an in-depth exploration of the phenomenon (qualitative aspect), and the quantitative analyses were supportive and exploratory rather than confirmatory. Future studies should aim for larger, more diverse samples, ideally drawn from multiple institutions or regions, to enhance generalizability and allow for more robust statistical analyses, including subgroup comparisons. Another limitation is the fact that there is no separate compulsory subject dedicated to cultural issues in the Polish undergraduate nursing education system. Knowledge in this area is taught piecemeal in other social science, basic, and specialised courses. However, the lack of a clear cultural emphasis may lead to difficulties in integrating this knowledge and putting it into practise. This limitation highlights the need for systemic changes in nursing curricula, advocating for dedicated, comprehensive modules on transcultural care. Future research could evaluate the impact of such curriculum changes on nurses’ cultural competence. Another limitation is that the study was conducted in a country with a relatively homogeneous cultural structure. Poland, as a largely ethnically homogeneous society, does not provide daily opportunities for contact with patients from different cultural backgrounds. This may influence the limited range of participants’ experiences and their difficulties in recognising and interpreting cultural phenomena in the context of healthcare. To mitigate this, future studies in similar contexts could incorporate qualitative methods that delve deeper into nurses’ limited experiences or explore the effectiveness of virtual reality or simulation-based training designed to expose nurses to diverse cultural scenarios. In addition, the study was conducted exclusively among graduate nursing students who, despite their work experience, are still functioning as learners. Although the principal investigator did not have direct contact with the participants, the influence of so-called ‘social pressure’ or the desire to give ‘correct’ answers cannot be completely excluded, especially as recruitment was conducted through lecturers. Future research should consider recruiting participants from a wider range of professional stages (e.g., experienced nurses, nursing educators) and employing recruitment strategies that further minimise potential social desirability bias. A final limitation is the use of a tool constructed on the basis of clinical case descriptions. Although this method allowed for the collection of rich qualitative material, it did not include actual observation of behaviour or objective assessment of competence in a clinical setting. Triangulation of data sources, e.g., through an analysis of simulation recordings or in-depth interviews, is worth considering in future research. This limitation can be addressed in subsequent studies by incorporating direct observational methods, simulated clinical encounters with diverse actors, or patient feedback to provide a more objective and comprehensive assessment of cultural competence in action.

## 5. Conclusions

The following conclusions can be drawn from the analyses:Cultural knowledge among the nurses surveyed is fragmentary and often limited to superficial aspects (e.g., language barriers), ignoring deeper cultural determinants (e.g., temporal orientation, territoriality, biological differences).Significant gaps in knowledge of issues such as non-verbal communication, cultural meaning of space, and specificity of health assessment in people with a different complexion were identified.The key findings of the study indicate that the cultural knowledge of the participants was fragmented and simplified, with clear deficits in the areas of non-verbal communication, biological differences, and understanding space in a cultural context. Three distinct profiles of cultural knowledge were identified: pragmatic, socio-reflective, and critical–experienced, highlighting individual differences in the perception and understanding of cultural phenomena in nursing care.Implications for nursing practice and education are as follows: These results have important implications for the design of nursing curricula. Instead of universal training, it is necessary to adapt educational content to the level of awareness, experience profile, and reflective readiness of nurses. Identifying knowledge profiles allows for the creation of targeted educational interventions that will more effectively fill knowledge gaps and develop specific competencies. For example, for students with a pragmatic profile, emphasis should be placed on theoretical foundations and awareness of cultural differences; for the profile with high social and communication sensitivity, it will be crucial to develop advanced intercultural communication skills and reflection on attitudes, while for the critical and experienced profile, training in strategies for dealing with complex barriers in care is recommended.

Our recommendations are as follows:
It is recommended that diagnostic tools (e.g., based on clinical scenarios) be implemented in the early stages of nursing education to identify students’ individual educational needs in terms of cultural competence.Training should be practice-oriented, using simulations, case studies, and field experience to equip nurses with specific knowledge of specific cultural dimensions (e.g., non-verbal communication, biological determinants of disease in different populations) and practical tools for effectively adapting care to patient needs (e.g., intercultural communication protocols, guidelines for pain assessment in different cultures). Instead of a general ‘equipping with tools’ approach, nurses should be taught specific techniques and strategies.It is advisable to conduct longitudinal studies evaluating the effectiveness of personalised educational programmes. It is also important to expand the research sample to include nurses with different professional experience and from different geographical regions as well as to include the patients’ perspective in the research in order to obtain a more complete picture of the quality of care in a multicultural environment. Further research should also explore the impact of actual clinical experience with patients from different cultures on the development of nurses’ knowledge and skills.

Although the current study employed a cross-sectional design, the inclusion of a longitudinal component or a comparison group could enhance the robustness of future research. We recommend that subsequent studies adopt a pre–post design or include a control group, such as students who have completed similar training, to better assess the effectiveness of educational interventions in transcultural nursing.

## Figures and Tables

**Figure 1 healthcare-13-01907-f001:**
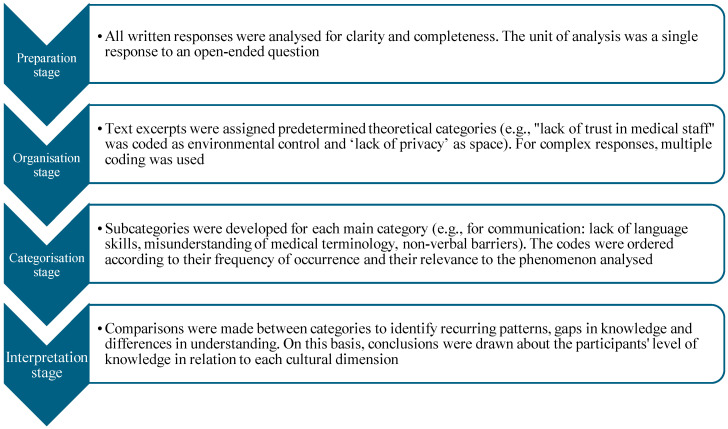
The coding process.

**Figure 2 healthcare-13-01907-f002:**
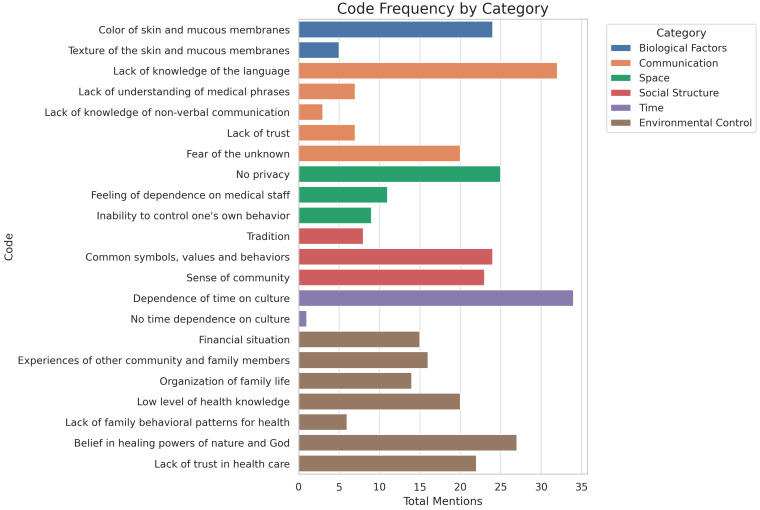
Code matrix.

**Figure 3 healthcare-13-01907-f003:**
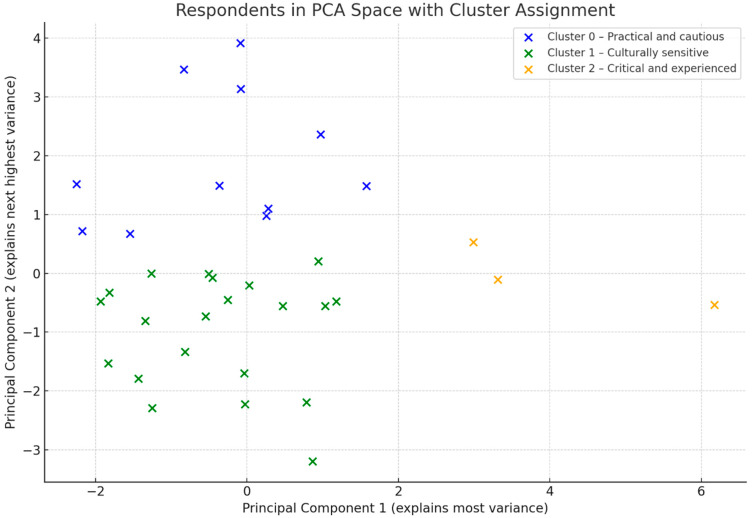
Principal component analysis (PCA) and distribution of respondents across clusters.

**Table 1 healthcare-13-01907-t001:** The code system adopted in the study.

1. Biological factors 1.1 Colour of skin and mucous membranes 1.2 Texture of the skin and mucous membranes
2. Communication 2.1 Lack of knowledge of the language 2.2 Lack of understanding of medical phrases 2.3 Lack of knowledge of non-verbal communication 2.4 Lack of trust 2.5 Fear of the unknown
3 Space 3.1 No privacy 3.2 Feeling of dependence on medical staff 3.3 Inability to control one’s own behaviour
4 Social structure 4.1 Tradition 4.2 Common symbols, values, and behaviours 4.3 Sense of community
5 Time 5.1 Dependence of time on culture 5.2 No time dependence on culture
6 Environmental control 6.1 Financial situation 6.2 Experiences of other community and family members 6.3 Organisation of family life 6.4 Low level of health knowledge 6.5 Lack of family behavioural patterns for health 6.6 Belief in the healing powers of nature and God 6.7 Lack of trust in healthcare

**Table 2 healthcare-13-01907-t002:** The frequency of codes in the six cultural dimensions of the Giger and Davidhizar model.

Cultural Phenomena Identified by Giger and Davidhizar	Segments	%
Environmental control	120	33.99
Communication	69	19.55
Social structure	55	15.58
Space	45	12.75
Time	35	9.92
Biological factors	29	8.22
Total	353	100

**Table 3 healthcare-13-01907-t003:** Statistical summary of codes.

Cultural Phenomena	Codes	Mean (M)	Standard Deviation (SD)
Biological factors	Colour of skin and mucous membranes	0.67	0.667
	Texture of the skin and mucous membranes	0.14	0.346
Communication	Lack of knowledge of the language	0.89	0.314
	Lack of understanding of medical phrases	0.19	0.396
	3 Lack of knowledge of non-verbal communication	0.08	0.276
	Lack of trust	0.19	0.396
	Fear of the unknown	0.56	0.497
Space	No privacy	0.69	0.517
	Feeling of dependence on medical staff	0.31	0.461
	Inability to control one’s own behaviour	0.25	0.433
Social structure	Tradition	0.22	0.478
	Common symbols, values, and behaviours	0.67	0.527
	Sense of community	0.64	0.480
Time	Dependence of time on culture	0.94	0.229
	No time dependence on culture	0.03	0.164
Environmental control	Financial situation	0.42	0.546
	Experiences of other community and family members	0.44	0.497
	Organisation of family life	0.39	0.541
	Low level of health knowledge	0.56	0.550
	Lack of family behavioural patterns for health	0.17	0.373
	Belief in the healing powers of nature and God	0.75	0.433
	Lack of trust in healthcare	0.61	0.487

**Table 4 healthcare-13-01907-t004:** Spearman’s rank correlation matrix.

Cultural Phenomena	Biological Factors	Communication	Space	Time	Social Structure	Environmental Control
Biological Factors	1.0	−0.05	−0.3	0.18	−0.16	0.01
Communication	−0.05	1.0	0.34	−0.24	0.17	−0.01
Space	−0.3	0.34	1.0	0.07	0.38	0.5
Time	0.18	0.24	0.07	1.0	0.14	0.19
Social Structure	−0.16	0.17	0.38	0.14	1.0	0.36
Environmental control	0.01	−0.01	0.5	0.19	0.36	1.0

**Table 5 healthcare-13-01907-t005:** Summary of cultural knowledge profiles.

Profile Name	Defining Characteristics	Dominant Categories	Knowledge Depth
Pragmatic	Focus on practical communication and biological needs; limited conceptual reflection.	Communication, Biological Factors	Basic/factual
Socio-Reflective	Emphasis on family, community, and social norms; includes moderate cultural sensitivity.	Social Structure, Environmental Control	Intermediate/contextual
Critical–Experiential	Demonstrates self-awareness, critical thinking, and contextualised understanding of diversity.	Environmental Control, Communication, Time	Advanced/reflective

## Data Availability

The data analysed in the study are available upon request to the first author.

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
