# Peer review of "Understanding Diversity: The Cultural Knowledge Profile of Nurses Prior to Transcultural Education in Light of a Triangulated Study Based on the Giger and Davidhizar Model"

_healthcare, 2025, doi:10.3390/healthcare13151907_

Round 1

Reviewer 1 Report

Comments and Suggestions for Authors

Comments to authors: healthcare-3722534

Introduction

  1. The introduction includes an extensive discussion of background information and theoretical models, but it lacks a clear narrative thread. The transitions between paragraphs are abrupt, and the rationale for choosing the Giger and Davidhizar model is buried late in the introduction. Please consider re-structuring the introduction using a funnel approach, begin broadly with the importance of cultural competence in nursing, then narrow down to the gap in research, the rationale for the chosen model, and finally, the specific aim and research question.
  2. Several points are repeated unnecessarily (e.g., cultural competence is not innate, requires learning, evolves through experience), which affects conciseness and reader engagement. Please eliminate repetition. Combine related sentences and streamline the prose for better flow.
  3. While several models are listed, the paper lacks a critical synthesis of why the Giger and Davidhizar model is the most appropriate for this study. Merely stating that it has "high practical utility" is not enough for scholarly justification. Please provide a brief comparison of the models and articulate the specific advantages of the Giger and Davidhizar model in assessing knowledge-based cultural competence among Polish nursing MSc students.
  4. The research aim and PICO question are introduced too late in the paragraph and feel somewhat disconnected from the rest of the narrative. Please state the research aim earlier, preferably after discussing the knowledge gap. The PICO question should then be presented as the operational framing of this aim, not as an isolated list.
  5. The concept of culture does not only refer to ethnicity, diverse cultural groups or religious beliefs, but also to everyday aspects of functioning in different environments, systems and organisational structures, such as gender, dress, gestures, cooking, meaning of food, perception of reality, thoughts, behaviours, attitudes and others.I suggest to briefly presenting the citation to support this statement.

Materials and Methods

  1. The explanation of triangulation is conceptually vague. The term "methodological triangulation" should be clarified—does it refer to data triangulation, method triangulation, or theory triangulation? Define triangulation type explicitly. Justify why it was necessary. Clarify how the quantitative and qualitative methods interacted or complemented each other.
  2. Excessively detailed for some aspects (e.g., semester schedule). Missing: Sampling strategy (purposive, convenience?), sample size justification, and potential for bias. Please focus on essential contextual details only (e.g., drop "summer/winter semester" info). Include a clear rationale for the chosen sample size and comment on generalizability. Acknowledge potential gender imbalance bias and volunteer bias.
  3. Ethical procedures (verbal vs. written consent) are confusing and inconsistent. It's unclear whether the instrument underwent validation beyond face/content validity. Please standardize the consent process: “Written informed consent was obtained…”.
  4. No mention of instrument validation (e.g., expert review, reliability testing for knowledge questions). The decision to analyze only “expertise” questions needs stronger justification.
  5. The coding process is sound but verbose and would benefit from a schematic or figure. The quantitative methods (ANOVA, Spearman, cluster analysis) are listed, but not clearly tied to research questions or hypotheses.
  6. Lacks clarity about verbal vs. written informed consent. No information about data storage, anonymity, or participant right to withdraw after data submission.

Results

  1. The use of "social control" instead of “social structure” in cluster analysis is inconsistent with earlier terminology based on the Giger and Davidhizar model.
  2. Tables are poorly formatted (e.g., misaligned values in Table 2), and some rows lack clarity (e.g., multiple entries running together). Figure 1 and Figure 2 are referenced but not described sufficiently.
  3. Redundant use of phrases like “high variation in responses confirms the uneven level of knowledge in this area” could be more precise.
  4. Some statistics are reported without exact p-values or effect sizes.
  5. The rationale for the number of clusters and method of selection is not provided. Please add a sentence explaining how the number of clusters was determined (e.g., elbow method, silhouette score) and clarify why k-means was chosen.

Discussion and limitations

  1. Some claims are broad and not sufficiently supported by data or citations. E.g., “Nurses' cultural knowledge is mainly based on theoretical sources rather than on actual clinical experience.”
  2. Although Polish context is discussed, broader implications for global nursing education are underdeveloped.
  3. While the need for differentiated training is noted, the recommendation remains general.
  4. This Discussion section is analytically sound and offers meaningful insights into the study’s findings. However, to meet the expectations of a high-impact journal, it requires improved structure, clearer writing, and deeper engagement with educational implications and international relevance.
  5. The limitations are merely mentioned at the end ("Limitations of the study") but not fully elaborating the solution or future recommendations to tackle these limitations.

Conclusion

  1. Certain points (e.g., "training should be personalized") are repeated unnecessarily.
  2. The section reads more like an extended discussion than a conclusion. There’s no clear separation between summary of findings, implications, recommendations, and future research.
  3. Phrases like “nurses should be equipped with the knowledge and practical tools…” are vague.

Author Response

Response to Reviewers

We thank the Reviewer for their valuable comments and constructive suggestions. Below we provide point-by-point responses. All changes made in the manuscript have been highlighted in yellow.

  1. The introduction includes an extensive discussion of background information and theoretical models, but it lacks a clear narrative thread. The rationale for choosing the Giger and Davidhizar model is buried late in the introduction.

Author Response: The introduction was restructured using a funnel approach. We began with the importance of cultural competence, reviewed models, and clearly presented the rationale for choosing the Giger and Davidhizar model. Redundancies were removed and coherence improved. Changes highlighted in yellow.

  1. The explanation of triangulation is conceptually vague. Define triangulation type explicitly. Justify why it was necessary.

Author Response: We clarified the use of methodological triangulation (method triangulation) and explained its relevance for credibility and convergence. Details were added and marked in yellow.

  1. Excessively detailed for some aspects (e.g., semester schedule). Missing: Sampling strategy, sample size justification, and potential for bias.

Author Response: We removed non-essential details and added information about purposive and convenience sampling. Sample size justification and potential gender/volunteer bias were acknowledged. Revisions are highlighted.

  1. Ethical procedures (verbal vs. written consent) are confusing. No information about data storage, anonymity, or right to withdraw.

Author Response: We confirmed written informed consent was obtained. Added statements on anonymity, secure data storage, and right to withdraw. See revised Ethics section in yellow.

  1. No mention of instrument validation. The decision to analyze only “expertise” questions needs stronger justification.

Author Response: We added that the tool underwent expert validation by three transcultural nursing specialists. The focus on expertise was justified in relation to cognitive self-assessment. Changes marked.

  1. The coding process is sound but verbose and would benefit from a schematic or figure.

Author Response: A schematic (Figure 1) illustrating the four-stage coding process based on Elo & Kyngäs was added. It clarifies the procedure and complements the description.

  1. Quantitative methods are listed, but not clearly tied to research questions or hypotheses.

Author Response: We expanded the explanation of the role of ANOVA, Spearman’s correlation, and cluster analysis in relation to the research question. Yellow highlights indicate changes.

  1. The use of 'social control' instead of 'social structure' is inconsistent.

Author Response: Terminology was corrected to 'social structure' throughout, in line with the Giger and Davidhizar model.

  1. Tables are poorly formatted and figures insufficiently described.

Author Response: Tables were reformatted for clarity and figures were supplemented with explanatory captions. Formatting now meets publication standards.

  1. Repetitive phrasing ('high variation…') could be more precise.

Author Response: Repetitive phrasing was refined for clarity and conciseness. See Results section for edits.

  1. Some statistics are reported without exact p-values or effect sizes.

Author Response: We included exact p-values and effect sizes (e.g., η²) where applicable.

  1. The rationale for the number of clusters and method of selection is missing.

Author Response: We added a sentence explaining the use of the elbow method and the rationale for selecting k-means clustering. See Data Analysis section.

  1. Discussion and conclusions lack depth and are repetitive.

Author Response: These sections were rewritten to better reflect findings, integrate implications, and reduce redundancy. Structural improvements are visible in yellow highlights.

  1. Limitations are too brief.

Author Response: We expanded the limitations section to acknowledge methodology constraints and outline directions for future research.

Reviewer 2 Report

Comments and Suggestions for Authors

The introduction is clear, well contextualized, and provides a good summary of the state of the art and theoretical foundation. I suggest specifying more clearly the knowledge gap this study aims to fill in order to strengthen its scientific contribution.

The use of methodological triangulation (quantitative and qualitative) is a strength. However, it is suggested to complement it with observational or longitudinal methods in future studies to enhance external validity.

Although there is a clear description of the instrument, procedure, and analysis, it would be useful to include more details about the content of the questions used and to justify why responses related to practical interventions were excluded.

The results are well organized, with clear statistical data and appropriate figures. A summary table of the identified profiles is recommended to facilitate understanding.

The conclusions adequately reflect the study’s findings. As a suggestion, the educational and policy implications could be emphasized more strongly.

Tables and figures are well designed, but it is recommended to add descriptions to improve visual accessibility.

Comments on the Quality of English Language

The English is understandable, but there are some grammatical constructions that sound unnatural and punctuation issues in certain sections. A language review by a native speaker would improve overall clarity.

Author Response

Response to Reviewer Comments

We thank the Reviewer for the constructive and thoughtful feedback. Below we provide point-by-point responses to each comment.
All changes made in the manuscript have been highlighted in yellow.

  1. Comment:

The introduction is clear, well contextualized, and provides a good summary of the state of the art and theoretical foundation. I suggest specifying more clearly the knowledge gap this study aims to fill in order to strengthen its scientific contribution.

Response:
Thank you for this suggestion. We have clarified the specific knowledge gap addressed by this study, namely, the lack of empirical data on the self-assessed cultural knowledge of nurses prior to formal transcultural training in Poland. This addition has been incorporated into the Introduction section and marked in yellow.

  1. Comment:

The use of methodological triangulation (quantitative and qualitative) is a strength. However, it is suggested to complement it with observational or longitudinal methods in future studies to enhance external validity.

Response:
We appreciate this insight. A recommendation to include observational or longitudinal methods in future research was added to the Methods section to acknowledge the value of enhancing external validity. The revision has been highlighted in yellow.

  1. Comment:

Although there is a clear description of the instrument, procedure, and analysis, it would be useful to include more details about the content of the questions used and to justify why responses related to practical interventions were excluded.

Response:
We expanded the description of the instrument to include further detail on the question content. We also explained that practical interventions were excluded to isolate the cognitive component of cultural competence. These revisions have been made in the Methods section and marked in yellow.

  1. Comment:

The results are well organized, with clear statistical data and appropriate figures. A summary table of the identified profiles is recommended to facilitate understanding.

Response:
Thank you for this recommendation. We have added Table 5, which summarizes the three identified knowledge profiles (Pragmatic, Socio-Reflective, and Critical-Experiential), including their defining characteristics and dominant categories. The table and its reference in the Results section are highlighted in yellow.

  1. Comment:

The conclusions adequately reflect the study’s findings. As a suggestion, the educational and policy implications could be emphasized more strongly.

Response:
We have strengthened the Conclusions section by adding a paragraph outlining the key educational and policy implications, particularly regarding curriculum planning and the personalization of transcultural learning. This addition is highlighted in yellow.

  1. Comment:

Tables and figures are well designed, but it is recommended to add descriptions to improve visual accessibility.

Response:
We agree with this suggestion. Descriptive captions have been added to the figures to improve clarity and accessibility. These updates are highlighted in yellow.

  1. Comment on English Language:

The English is understandable, but there are some grammatical constructions that sound unnatural and punctuation issues in certain sections.

Response:
A thorough language revision has been completed to improve clarity, grammar, and punctuation. We also acknowledge that a final review by a native English speaker would further enhance the quality.

Reviewer 3 Report

Comments and Suggestions for Authors

Thank you for the opportunity to review this manuscript, which I read with interest. I did struggle with the paper overall because it focuses on cultural knowledge, which is only one useful element within cultural competence. I find cultural safety more helpful - a term initiated by a nurse, also in the 1970s - and cultural humility is also a good one.  'competence' is not easily assessed and it should be assessed by the patient. It is not so much about displaying cultural knowledge as it is about treating people with respect and dignity, and curiosity about the things that are important to them. Knowledge of culturally-specific communication such as gestures may increase a nurse's ability to perform in culturally-appropriate ways but it also begs a movement away from authenticity and towards performance. I would warn against this.

It might be worth nodding to this complexity in your discussion.

In terms of providing empirical evidence to support an existing model, this paper does a good job of that.

Table 3 has a typo - cods

thanks and best wishes

Author Response

Response to Reviewer

We sincerely thank the Reviewer for the insightful and thought-provoking feedback, which we greatly appreciated.

Reviewer comment:
“I did struggle with the paper overall because it focuses on cultural knowledge, which is only one useful element within cultural competence... I find cultural safety more helpful... and cultural humility is also a good one... ‘Competence’ is not easily assessed and it should be assessed by the patient... I would warn against this.”

Author response:
We fully agree that cultural knowledge, while valuable, represents only one dimension of what it means to provide culturally safe and responsive care. In response to your suggestion, we have revised the Discussion section to acknowledge the complexity of cultural competence and reflect on the importance of concepts such as cultural safety and cultural humility, both of which have shaped nursing theory and practice since the 1970s. We emphasize that while our study focused on cognitive self-assessment, cultural competence should be critically examined as a dynamic and relational construct, ultimately judged by the patient. These additions are marked in yellow in the manuscript.

Reviewer comment:
“Table 3 has a typo – cods”

Author response:
Thank you for noting this. The typo in Table 3 (“cods”) has been corrected to “codes.” The change is highlighted in yellow.

Reviewer 4 Report

Comments and Suggestions for Authors

in attachment

Author Response

Response to Reviewer

We thank the Reviewer for their thoughtful and detailed comments. Below we provide our responses to each point raised.
All changes made to the manuscript have been highlighted in yellow.

Comment 1:

The Giger and Davidhizar Transcultural Assessment Model is well-established, practical, and widely used in nursing, lending credibility and structure to the study. The focus on MSc nursing students with professional experience ensures participants have relevant clinical exposure, which strengthens the reliability of their self-assessment. But self-reported data can be subjective and biased; participants may overestimate or underestimate their knowledge and competence. If participation was voluntary, there may be a selection bias—those already interested in cultural issues may be more likely to participate. The sample is restricted to Polish MSc nursing students, which may limit the applicability of findings to other countries or educational contexts.

Response:
We fully agree with this important observation and have addressed it in the Limitations section. A new paragraph was added to acknowledge the potential limitations of self-assessment, including subjectivity and possible over- or underestimation of cultural knowledge and competence. We also recognized the possibility of selection bias due to voluntary participation, as well as the limited generalizability of the findings given the cultural and geographic specificity of the Polish sample. These revisions have been marked in yellow in the manuscript.

Comment 2:

No longitudinal component: A pre-post design (before and after the multicultural course) could provide stronger evidence of training impact. No comparison group: Including a control group (e.g., students who had completed similar training elsewhere) would have improved the strength of conclusions.

Response:
We appreciate this valuable suggestion. While the present study used a cross-sectional design, we acknowledged in the Recommendations section that including a longitudinal element or a comparison group would enhance the strength of future studies. We suggest that future research adopt a pre-post design or include a control group (e.g., students who have completed similar training elsewhere) to better assess the impact of educational interventions. This has been added to the manuscript and highlighted in yellow.